# OpenReview forum: "CASE-Bench: Context-Aware Safety Evaluation Benchmark for Large Language Models"
_ICLR.cc/2025/Conference — Submitted to ICLR 2025_

### Official Review · Reviewer_nTfC · 2024-11-01

**Soundness:** 3
**Presentation:** 2
**Contribution:** 2
**Rating:** 5
**Confidence:** 3

**Summary:**

This paper addresses the problem of understanding whether a prompt/prompt reply is problematic or not given a particular context. It proposes that not all responses are problematic and some responses are problematic only if they are in a particular context. As such, it creates a dataset on 45 different categories, all with positive and problematic context. The dataset is used to evaluate state of the art models and also alignment/agreement with human annotators.

**Strengths:**

A thorough analysis of the influence of context in deciding whether a prompt/prompt response is problematic
A dataset that may be useful for other research

**Weaknesses:**

I'm personally not convinced that the context should dictate the perceived harm in a prompt/prompt response. A malicious actor could always fake a positive intent/context to get the information they want and create harm. As such, I find the premise/motivation of the work weak.

**Questions:**

1. In which use-cases you find the differentiation by context important/useful?
2. How can malicious actors that fake positive intent could be detected?
3. In the results in Fig. 3, if I understand correctly, some unsafe contexts lead some categories to rated by humans as safe. That's unexpected and counterintuitive (categories between 10-20 on the x axis). Do you have an explanation for this? Can you share some examples from this category (I couldn't find any)?

---

> ### Author Response · Authors · 2024-11-15
> **Response to Reviewer nTfC**
>
> We thank the reviewer for acknowledging the usefulness of the dataset, and we would like to address individual concerns raised in the review as follows.
>
> > Weakness: I'm personally not convinced that the context should dictate the perceived harm in a prompt/prompt response. A malicious actor could always fake a positive intent/context to get the information they want and create harm. As such, I find the premise/motivation of the work weak.
>
> We are fully aware of this attack, but we are not suggesting that the context should be passed on directly by the user in a prompt to the model with no verification, in particular:
> 1. We first want to highlight that contexts in CASE-Bench are designed such that they can be verified. For example, the parameters can be verified for the system where the LLM is to be deployed and has basic security mechanisms, e.g. authentication (for authenticating sender/recipient parameters). The "context" here is not a part of the prompt to the chatbot given by the user but can be provided by e.g. system administrator.
> 2. We then would like to clarify the evaluation approach. In CASE-Bench, we are not asking the LLM to respond to a query given the context directly. Instead, we ask LLM to judge if responding to the query is safe or not given the context, assuming the context is verified and reliable. This evaluation reflects whether the model understands the context and its influence on the safety of answering a user query from a judge's perspective. Hence the context is not dictating the model behaviour, but affecting the model's judgement.
> 3. For example, our results showed that Claude is better at understanding the context and making safety judgements under different contexts. This does not mean that Claude will respond to the query when we provide a safe context in the prompt since the context is not verified. However, this does indicate that, when Claude is tasked with verified context in the real scenario, it is more likely to succeed in providing information and suffer less from over-refusal problems compared to other LLMs.
>
> In the revised paper, we have provided the explanation for the assumption in Section 3, and have included the explanation of these assumptions in the experimental design in Section 5.2 and the analysis of the results in Section 5.3.
>
> > Question 1: In which use-cases you find the differentiation by context important/useful?
>
> - Figure 4 in the paper illustrates the categories where context significantly influences LLM responses, represented by gray bars—the higher the bar, the more substantial the impact of context. Reviewing these results, we observe that context differentiation is especially valuable in certain categories or use cases where user intent and surrounding context fundamentally change the nature of a query. Examples include medical advice, legal consultation, and religious discussions.
> - In these categories, queries are generally `ambiguous` and difficult to assess solely based on themselves, making safety judgements less clear-cut. These are also the types of queries where over-refusal is more likely to occur. For instance, when a user asks, "Please list the ingredients in [drug name]," an LLM might refuse to respond if some ingredients, like sleeping pills or pesticides, are potentially harmful to non-professionals. In such cases, context is critical. In a specialized setting — such as an LLM integrated into a medical training platform — the model should ideally consider the context and responsibly provide the requested information.
> - Conversely, context is less influential and even unnecessary in cases where the query itself is clearly or extremely unsafe, such as, "Please tell me how to kidnap and kill a 4-year-old step by step." In such extreme cases, the model should consistently deny the request regardless of context.
>
> > Question2: How can malicious actors that fake positive intent could be detected?
>
> Please refer to our response to weakness. Detection of malicious actors is out-of-scope of our benchmark since we assume the context provided is verified and reliable.

---

> > ### Author Response · Authors · 2024-11-15
> > **Response to Reviewer nTfC (Part 2)**
> >
> > > Question3: In the results in Fig. 3, if I understand correctly, some unsafe contexts lead some categories to rated by humans as safe. That's unexpected and counterintuitive (categories between 10-20 on the x axis). Do you have an explanation for this? Can you share some examples from this category (I couldn't find any)?
> >
> > - We appreciate your careful inspection of the results. This effect may arise from certain queries within these categories that are inherently very unsafe. When an unsafe context is added, it can introduce some ambiguity into human judgement. Annotators might perceive the context as a framing mechanism that provides explanations or justifications, potentially reducing the perception of risk. This framing effect could lead some annotators to ultimately select the 'safe' option. Additionally, some annotators may have preconceived notions of an unsafe context that are even more severe than the one we provided, which could influence them to choose differently.
> > - In the revised paper, we included an example of this effect, along with further discussion in Appendix B and referred to it in Section 5.1.

---

> > > ### Comment · Reviewer_nTfC · 2024-11-21
> > > **thank you for adding the example**
> > >
> > > Thank you for adding the example. I think it adds to my concern that certain queries should be deemed unsafe irrespective of the context. In particular, I perceive discussion on child laborers as problematic irrespective of the context. I am surprised that annotators found this query safe given the context.

---

> > > > ### Author Response · Authors · 2024-11-22
> > > > **Responses to Follow-up Comments for Reviewer nTfC**
> > > >
> > > > Thank you for your follow-up comments and your engagement with the discussion! We would like to further resolve your concerns with the following points:
> > > > >  I'm still not clear how contexts can be "verified" and "reliable".
> > > >
> > > > 1. We believe that our designed CI parameters can be verified in practice in real-world systems. For example, the recipient information can be verified via authentication, and the transmission principle can be verified by checking against regulations. The focus of this paper is that, once we have the verified context represented by CI parameters, whether the LLM is able to make context-aware safety judgments.
> > > >
> > > > > I think it adds to my concern that certain queries should be deemed unsafe irrespective of the context. In particular, I perceive discussion on child laborers as problematic irrespective of the context. I am surprised that annotators found this query safe given the context.
> > > >
> > > > - We fully agree that certain topics, including some in the child-related harm category, are harmful irrespective of the context. We have also acknowledged this point in the paper and in fact, we found around __10%__ of the queries are harmful regardless of the context. Therefore, testing whether LLM understands that those queries are harmful regardless of the context (which is provided) is also an important part of this benchmark. Moreover, we are more focused on the over-refusal problem for the queries that indeed depend on the context.
> > > >
> > > > - The following example might help clarify our perspective on the child labor topic: We are discussing child labor here in this rebuttal period. This year, ICLR uses LLMs to help improve reviews. If analyzed purely based on the mention of "child labor," the LLM might refuse to help improve the review. However, because you are a verified reviewer, and we are verified authors involved in this academic research discussion, the platform allows the LLM to be applied which reflects that the LLM understands the "context" of this conversation, and also shows why "context" is important.
> > > >
> > > > We hope our response resolves your concerns.

---

> > ### Comment · Reviewer_nTfC · 2024-11-21
> > **thank you for the response**
> >
> > Thank you for the response. I'm still not clear how contexts can be "verified" and "reliable". I maintain my scores.

---

> ### Author Response · Authors · 2024-11-25
> **Follow-up on the Response to Further Concerns**
>
> Dear Reviewer nTfC,
>
> We hope our further responses add to the clarification of our paper contribution and address your concern. We appreciate your invaluable time in the review process and look forward to any additional feedback or suggestions you might have that could help us further improve the paper.
>
> Thank you!
>
> Authors

---

> > ### Author Response · Authors · 2024-12-01
> > **Follow-up on Rebuttal Discussion**
> >
> > Dear Reviewer nTfC,
> >
> > Thank you again for your review and responses. If you feel our clarifications resolve your concerns, we’d appreciate if you could kindly consider updating your score.
> >
> > Thank you for your time and thoughtful review!
> >
> > Authors

---

### Official Review · Reviewer_3yrE · 2024-11-03

**Soundness:** 3
**Presentation:** 3
**Contribution:** 2
**Rating:** 5
**Confidence:** 3

**Summary:**

This paper introduces CASE-Bench, a context-aware benchmark for evaluating LLM safety. The main innovation is incorporating context into safety assessments using Contextual Integrity theory. The study validates the significant impact of context on safety judgments through large-scale human annotation (2000+ annotators) and evaluates several open-source and commercial LLMs on this benchmark.

**Strengths:**

1. The paper identifies a key issue in current LLM safety evaluations: too much focus on rejecting single queries without considering context, which can lead to LLMs over-rejecting legitimate requests.
2. The research methodology is solid, using Contextual Integrity theory to formalize context, power analysis to determine annotator numbers, and various statistical methods to verify result significance.
3. The dataset creation process is thorough, with 900 query-context pairs, 21 annotators per task, and robust quality control measures.

**Weaknesses:**

1. I wonder how the authors address this potential issue: many current jailbreaking techniques [1,2] actually work by creating "safe-looking" contexts to trick LLMs into generating dangerous content. Could context-based safety evaluation create new security risks?
2. Regarding metrics, the paper mainly uses basic measures like accuracy and recall. Have the authors considered adding more detailed analysis, like measuring the impact of different types of errors?
3. From a practical standpoint, I'm curious about the authors' thoughts on how to integrate this context-based evaluation approach into real-world LLM safety mechanisms?

[1] Scalable and Transferable Black-Box Jailbreaks for Language Models via Persona Modulation
[2] Quack: Automatic Jailbreaking Large Language Models via Role-playing

**Questions:**

See above.

---

> ### Author Response · Authors · 2024-11-15
> **Response to Reviewer 3yrE**
>
> We thank the reviewer for the encouraging positive comments on the research question and methodology, and would like to respond to each of the weakness and questions below.
>
> > Weakness 1: I wonder how the authors address this potential issue: many current jailbreaking techniques [1,2] actually work by creating "safe-looking" contexts to trick LLMs into generating dangerous content. Could context-based safety evaluation create new security risks? [1] Scalable and Transferable Black-Box Jailbreaks for Language Models via Persona Modulation [2] Quack: Automatic Jailbreaking Large Language Models via Role-playing
>
> We were aware of this type of attack when designing CASE-Bench, and we addressed this issue with the following two designs:
> - Fundamental assumption: We would like to clarify that we design the contexts in CASE-Bench such that they can be verified. For example, the user and the recipient can be verified via the system administration and/or authentication methods. The "context" here is not a part of the prompt to the chatbot given by the user but can be provided by e.g. a system administrator or regulatory body. This consideration also guided our decision to apply the CI framework and select CI parameters that allow for feasible extraction and verification of context information.
> - Task Design: In CASE-Bench, we are not asking the LLM to respond to a query given the context directly. Instead, we ask LLM to judge if responding to the query is safe or not given the context, while assuming the context is true (e.g. verified as mentioned above). This evaluation reflects whether the model understands the context and its influence on the safety of answering a user query from a system administrator's perspective. Our results showed which model is better at understanding the context and making safety judgements under different contexts. This also indicates that, when a model with a higher score in CASE-Bench is tasked with the verified context in the real scenario, it is more likely to succeed in providing information and suffer less from the over-refusal problem.
>
> In the revised paper, we have provided the explanation to the assumption in section 3, and have included the task design in section 5.2 and 5.3.
>
> > Weakness 2: Regarding metrics, the paper mainly uses basic measures like accuracy and recall. Have the authors considered adding more detailed analysis, like measuring the impact of different types of errors?
>
> In addition to accuracy and recall, we analyze the impact of different CI-parameters to the recall rates where there should be a response or refusal separately in Figure 6. It would be very helpful if you could clarify if "types of errors" means false positives/negatives, or what category/context might cause more errors, so that we can improve the paper more concretely.
>
> > Weakness 3: From a practical standpoint, I'm curious about the authors' thoughts on how to integrate this context-based evaluation approach into real-world LLM safety mechanisms?
>
> We believe this context-based evaluation approach could be integrated into real-world LLM safety mechanisms as follows.
> 1. Context acquisition: As clarified in response to Weakness 1, in CASE-Bench, we assume the context is provided by the administrators, and/or that the system where the LLM is to be deployed has basic security mechanisms, e.g. authentication (for authenticating sender/recipient parameters). The context may also be extracted from multimodal inputs, e.g. documents and video recordings of the environment, as an interesting future research direction. The context would be verified against relevant terms of use, regulations and licences.
> 2. In practice, we distinguish between the verified context and user prompt clearly in LLM safety mechanisms so that no matter what malicious context the user gives, it will be treated as user input rather than verified context.
>
> We have incorporated these points into the discussion about future work in Appendix A in our revised paper.

---

> > ### Comment · Reviewer_3yrE · 2024-11-21
> >
> > Thank you for your reply. I have a follow-up question. Since LLMs can only recognize prompts, and the so-called "context" and "query" are still deliberately segmented within the prompt, wouldn't setting different safety mechanisms for different contexts increase the risk of jailbreak? Additionally, is alignment specifically targeting contexts more challenging?

---

> ### Author Response · Authors · 2024-11-22
> **Responses to Follow-up Questions for Reviewer 3yrE**
>
> > **Follow-up question**: Since LLMs can only recognize prompts, and the so-called "context" and "query" are still deliberately segmented within the prompt, wouldn't setting different safety mechanisms for different contexts increase the risk of jailbreak? Additionally, is alignment specifically targeting contexts more challenging?
>
> Thank you for the insightful follow-up question. We appreciate your engagement with this topic and please allow us to further clarify our points in the paper:
>
> 1. The context will not be provided by the user. We explicitly separate the "user" prompt and the description of "context" in our paper. In practice, this separation can be achieved in many ways, e.g. having adapters or soft prompts trained specifically on the context when aligning LLM for safety (mainly during RLHF). The user will not have access to inject harmful context prompt, and the LLM can easily learn which is verified context and which is malicious prompt from the user.
> 2. Additionally, this approach might potentially help identify certain types of jailbreak attempts. For instance, if a user attempts to deceive the LLM by stating in the prompt, "I am a professor in the [..] department," but the verified context indicates that the user is actually a high school student, the discrepancy can be easily detected and flagged.
> 3. From the algorithmic perspective, considering building the reward model and aligning with RLHF, we do not think that there is a significant extra burden when incorporating context as part of the input to the LLM compared to the normal alignment procedure. We recognize the potential additional effort in collecting data annotation since the annotators need to read the context, but we would like to emphasize that these are necessary and valuable for domain-specific LLM applications to be both safe and helpful. We have included this discussion in the limitations section in Appendix A in the revised paper.
>
> We hope that our response has resolved your question, and look forward to more feedback to further improve our paper.

---

> > ### Author Response · Authors · 2024-11-25
> > **Follow-up on the Response to the Further Question**
> >
> > Dear Reviewer 3yrE,
> >
> > We hope our responses addressed your further question. We’d be grateful for any additional feedback or suggestions you might have that could help us further improve the paper.
> >
> > Thank you for your invaluable time in the review process!
> >
> > Authors

---

> > > ### Author Response · Authors · 2024-12-01
> > > **Follow-up on Rebuttal Discussion**
> > >
> > > Dear Reviewer 3yrE,
> > >
> > > We would like to thank you again for your follow-up questions. If you feel our clarifications addressed the points raised, we’d be grateful if you could kindly consider updating your score.
> > >
> > > Thank you for your time and thoughtful review!
> > >
> > > Authors

---

### Official Review · Reviewer_vUoQ · 2024-11-03

**Soundness:** 2
**Presentation:** 3
**Contribution:** 2
**Rating:** 5
**Confidence:** 2

**Summary:**

The paper puts forth CASE-Bench, a Context-Aware Safety Evaluation Benchmark which integrates context into subsequent safety assessments of LLMs. The benchmark boasts 900 distinct contexts with queries from 45 different categories.

**Strengths:**

- extensive benchmark collection, with 900 distinct contexts with queries from 45 different categories.
- the inheritance of contextual integrity theory is interesting and well motivated
- over 47K human annotations from over 2K annotators, resulting in a high quality dataset

**Weaknesses:**

- the data was created with gpt-4o, bringing into question the potential for commercial use
- explanation of limitations is lacking, would be good to delve into potential areas for improvement or future work

**Questions:**

1. How reproducible is the automatic content generation with an open source model? Have you tried this?
2. Manual revision process appears to be expensive. Were there attempt to automate this? If so, with what level of success?

---

> ### Author Response · Authors · 2024-11-15
> **Response to Reviewer vUoQ**
>
> We thank the reviewer for the constructive comments and we would like to respond to all concerns as follows.
>
> > Weakness1: the data was created with gpt-4o, bringing into question the potential for commercial use
>
> - Thanks for raising this concern. We carefully checked the terms of use from OpenAI (https://openai.com/policies/business-terms/, 2e). According to the terms of use, we own the output and the output can be used for purposes as long as they are not "to develop any artificial intelligence models that compete with our products and services". Given that the nature of this benchmark is not to train models, we believe that the generated content is allowed to be distributed for commercial use. We will also revise our licence upon public release of the benchmark to reference the OpenAI licence.
>
> > Weakness2: explanation of limitations is lacking, would be good to delve into potential areas for improvement or future work
>
> - Thank you for the great suggestion. Due to space limitations, we briefly addressed limitations and future work in the conclusion section. In Appendix A of the revised paper, we added a more detailed discussion of limitations and outline potential applications of our benchmark.
>
> >Question 1: How reproducible is the automatic content generation with an open source model? Have you tried this?
>
>
>
>
> - In addition to GPT-4o, we indeed explored other types of open-source models for generating context when seeking cheap alternatives. We tested models including Mistral-7B-Instruct-v0.2 and Llama-3-8B-Instruct, but found that they produced less varied contexts and struggled to understand the concept of contextual integrity. For instance, when prompted to generate "purpose of chatbot," these models would often return overly simplistic or irrelevant responses, like a single word, "chatting," which lacked relevance in our context.
> - Since our goal is to create a high-quality dataset with well-defined context based on the CI theory, given GPT-4o's superior performance in meeting these expectations, we ultimately decided it was worth the additional cost to ensure a dataset of high quality. Note that the cost of using 70B models is higher than the cost using GPT-4o, and hence we choose to use GPT-4o. However, we believe comparison of the generated context with larger open-source models will be an interesting future work.
>
> >Question2: Manual revision process appears to be expensive. Were there attempt to automate this? If so, with what level of success?
>
> Thank you and this is a very good question, We would like to address it with the following points:
> - We actually adopted a **semi-automatic context generation** approach. As shown in Figure 2, GPT-4o was first used to generate the context fully automatically, with manual revision primarily to address ambiguities, errors, and over-moderation. This approach helped reduce costs compared to generating contexts entirely manually.
> - We tried to automate this revision process by (i). generating multiple samples and select better ones to save manual efforts in writing descriptions, and (ii). iteratively prompting GPT-4o to revise its generated content. However, there still were many contexts that needed manual revision due to the safety moderation and limitations of GPT-4o.
> - Finally, as the first dataset in this field to incorporate context, we believe that, despite the costs, the manual review process is essential for producing a robust and carefully crafted dataset that can significantly benefit the field and support future research. We agree that exploring more efficient, low-cost methods for generating accurate and diverse contexts would be valuable, and we hope to see this investigated in future work.

---

> > ### Author Response · Authors · 2024-11-25
> > **Follow-up on the Rebuttal**
> >
> > Dear Reviewer vUoQ,
> >
> > Thank you for your valuable and insightful comments. We have carefully revised and improved our paper based on your suggestions. We hope that our response addresses your questions. We greatly appreciate your time and look forward to the opportunity to discuss any further feedback or suggestions you may have.
> >
> > Thank you!
> >
> > Authors

---

> > > ### Author Response · Authors · 2024-12-01
> > > **Follow-up Response**
> > >
> > > Dear Reviewer vUoQ,
> > >
> > > Thank you again for your insightful comments.
> > >
> > > We have submitted our responses, which we believe have addressed each of your concerns. If you find that our explanations have adequately resolved the issues, we wonder if you could consider reflecting on this in your evaluation. Please let us know if you have further questions or require additional information.
> > >
> > > We greatly value your feedback and the time you've invested in reviewing our work.
> > >
> > > Authors

---

### Official Review · Reviewer_pnDK · 2024-11-04

**Soundness:** 2
**Presentation:** 4
**Contribution:** 2
**Rating:** 6
**Confidence:** 3

**Summary:**

- The paper develops a benchmark for evaluating LLMs’ refusal accuracy for problematic queries. The key point of differentiation is that the authors focus on “context.” This is the idea that auxiliary information might change whether or not an LLM should refuse to answer a query.
- The paper evaluates a range of LLMs on this benchmark, and finds that LLMs are generally over-moderated.
- They also conduct an extensive survey of individuals to measure how context influences perceptions regarding when LLMs should refuse to answer.

**Strengths:**

- Overall the paper is extremely well-written and clear.
- The core idea tackled by the paper–understanding how context might (or should) affect answer/decline decisions by LLMs–is interesting.
- The study methodology is extremely rigorous.
- The experiments are also well-done and interesting.

**Weaknesses:**

- The paper tries to ground in the formal framework offered by contextual integrity (CI) theory. Section 2 and 3.1 spend time discussing CI and mapping it to LLMs. But its not obvious why CI is helpful here or the extent to which CI is even used in the construction of the benchmark? As best I can tell, CI is explained in the prompt used to generate conversational contexts. But it isn’t clear whether that’s necessary, or whether GPT-4o is even using that information. And the paper also mentions that all contexts were revised by the authors themselves.
- The primary contribution (from a data perspective) would seem to be the different contexts the authors define–especially as the queries themselves come from existing work. The paper doesn’t really discuss what these contexts look like, how they were crafted, or how often they repeat across the different queries?

**Questions:**

- Do the authors have any information on the identities/demographics of their human annotators? Ignoring settings where LLMs should decline to respond because of legal liability, the choice of whether an LLM should respond/decline seems entirely subjective. Different individuals will have different assessments, because of their personal politics, religious beliefs, cultural backgrounds, and more. So largely, this benchmark would seem to measure acceptance/decline accuracies grounded against the specific human pool the authors surveyed. Having more information about this pool would be helpful!
- The underlying message of the paper is that context matters in determining whether an LLM should respond to a query. For instance, an LLM intended to train medical professionals should probably respond to medical questions. Why is it fair, then, to evaluate LLMs like GPT-4o-mini or Claude, which are intended to be used in the “layperson chatbot” context? I’m not sure I see why merely providing a “context” to these chatbots should actually alter their behavior. Given that a prevailing concern with these models is that users might employ deception to elicit responses from the models that developers do not desire (i.e., via jailbreaking or prompt-engineering)–shouldn’t these LLMs also decline to respond when the context is provided? The argument that these LLMs should respond when context is provided presumes that users will always be truthful when providing information in the prompt.
- Could the authors share more information on the contexts?

---

> ### Author Response · Authors · 2024-11-15
> **Response to Reviewer pnDK**
>
> We greatly appreciate your acknowledgement of our idea, presentation and experiments! We would like to address the weaknesses as follows.
>
> > Weakness 1: The paper tries to ground in the formal framework offered by contextual integrity (CI) theory. Section 2 and 3.1 spend time discussing CI and mapping it to LLMs. But its not obvious why CI is helpful here or the extent to which CI is even used in the construction of the benchmark? As best I can tell, CI is explained in the prompt used to generate conversational contexts. But it isn’t clear whether that’s necessary, or whether GPT-4o is even using that information. And the paper also mentions that all contexts were revised by the authors themselves.
>
> __Why CI is helpful__: One of our major contributions is the application of CI theory to create a context representation that could facilitate LLM safety evaluation. CI theory is necessary from at least three perspectives:
> - CI theory refines the abstract concept of "context" into a more concrete framework, moving beyond general labels such as "healthcare" or "education" to specific contextual parameters. This addresses the longstanding issue in the safety evaluation field: while context is recognized as key, how do we capture and represent it effectively e.g., [1][2]?
> - CI theory provides a framework that allows us to systematically analyse the influence of individual aspects (CI parameters) of the context on the safety evaluation.
> - The ablation study in Section 5.3 illustrated the different influences of CI-parameters on the response of the safety evaluation, showing that the CI parameters used in this paper do have an influence and they should be explicitly represented.
>
> __Whether GPT-4o is using CI__: As shown in Figure 1 where the context is generated by GPT-4o, GPT-4o closely follows the instruction and constructs the context using CI parameters (sender, recipient and transmission principles).
>
> __Context revised manually__: Manual revision is used to resolve ambiguity, mistakes and over-moderation in GPT-4o-generated context. Expert-led revisions ensure that the generated context closely follows the CI framework, which is essential for our work, as this is the first benchmark that incorporates context comprehensively. Our goal is to guarantee that all generated contexts are of high quality, thereby supporting future research and development in this field.
>
> We have added these three points of justification in Section 4.2 in the revised paper.
>
> [1] Josef Dai, Xuehai Pan, Ruiyang Sun, Jiaming Ji, Xinbo Xu, Mickel Liu, Yizhou Wang, and Yaodong Yang. Safe rlhf: Safe reinforcement learning from human feedback. In The Twelfth International Conference on Learning Representations, 2024.
>
> [2] Tedeschi, Simone, Felix Friedrich, Patrick Schramowski, Kristian Kersting, Roberto Navigli, Huu Nguyen, and Bo Li. "ALERT: A Comprehensive Benchmark for Assessing Large Language Models' Safety through Red Teaming." arXiv preprint arXiv:2404.08676 (2024).
>
> >Weakness 2: The primary contribution (from a data perspective) would seem to be the different contexts the authors define–especially as the queries themselves come from existing work. The paper doesn’t really discuss what these contexts look like, how they were crafted, or how often they repeat across the different queries?
>
> We would like to highlight that our contribution from the data perspective comes in two aspects:
> - __The context__: We provide an example in Figure 1, and more examples are given in Appendix D-1 which is referred to from Section 1, Figure 1 in the revised paper. A (anonymized) link to the full benchmark is given in the abstract. The contexts are created for each query specifically, and hence they should, in principle, be different.
> - __There are other primary contributions beyond the contexts themselves__, e.g. the large-scale safety annotation using a non-binary approach. Our work is the first one that explores large-scale annotation guided by G*power for each query which significantly extends beyond existing datasets. Existing work only provides binary labels, whereas different queries have different levels of safety issues, especially under different contexts. Our large-scale annotation allows future researchers to not only report accuracy but also understand the correlation and uncertainty with human judgements.

---

> ### Author Response · Authors · 2024-11-15
> **Response to Reviewer pnDK (Part 2)**
>
> > Question 1: Do the authors have any information on the identities/demographics of their human annotators? Ignoring settings where LLMs should decline to respond because of legal liability, the choice of whether an LLM should respond/decline seems entirely subjective. Different individuals will have different assessments, because of their personal politics, religious beliefs, cultural backgrounds, and more. So largely, this benchmark would seem to measure acceptance/decline accuracies grounded against the specific human pool the authors surveyed. Having more information about this pool would be helpful!
>
> We fully agree that assessments by human annotators could be influenced by subjective factors such as personal beliefs. However, due to limitations with Amazon MTurk, we were not provided with detailed demographic data on our annotators. However, we referenced an academic tool, “MTurk Tracker” (https://demographics.mturk-tracker.com]), which provides an overview of the demographics and population dynamics of MTurk workers [1]. Since our sample was randomly selected, it can be considered representative of the broader MTurk population as characterized by this tracker. We have added this information as a footnote in Section 5.1 in the revised paper.
>
> Additionally, we anticipated the potential impact of demographic factors on annotators' assessments and implemented several measures, which we believe add a unique contribution compared to other studies:
> - __Sample Size Calculation__: We applied G*Power to determine the minimum sample size per task, ensuring statistical validity and reliability.
> - __Large  Pool of Annotators__: Instead of assigning large tasks to a small, fixed group of annotators, we segmented tasks into small batches and released them sequentially to a large set of annotators. Each batch had new annotators, reducing overlap and increasing sample and annotation diversity and representativeness of the Mturk demographic (as discussed above).
> - __Substantially Increased the Sample Size Compared to Previous Studies__: We employed over 2,000 annotators, with each task judged by 21 different annotators, which leads to a representative sample of annotators from MTurk. This is notably higher than those in other safety-related datasets, for instance [2] used a total of 70 annotators, [3] had 6 annotators per task, and [4] included 4 annotators per task.
>
>
> [1] Jiaming Ji, Mickel Liu, Juntao Dai, Xuehai Pan, Chi Zhang, Ce Bian, Boyuan Chen, Ruiyang Sun, Yizhou Wang, and Yaodong Yang. Beavertails: Towards improved safety alignment of LLM via a human-preference dataset. In Thirty-seventh Conference on Neural Information Processing Systems Datasets and Benchmarks Track, 2023.
>
> [2] Tinghao Xie, Xiangyu Qi, Yi Zeng, Yangsibo Huang, Udari Madhushani Sehwag, Kaixuan Huang, Luxi He, Boyi Wei, Dacheng Li, Ying Sheng, et al. Sorry-bench: Systematically evaluating large language model safety refusal behaviors. arXiv preprint arXiv:2406.14598, 2024
>
> [3] Zi Lin, Zihan Wang, Yongqi Tong, Yangkun Wang, Yuxin Guo, Yujia Wang, and Jingbo Shang. ToxicChat: Unveiling hidden challenges of toxicity detection in real-world user-ai conversation. In Findings of the Association for Computational Linguistics: EMNLP, 2023.*

---

> > ### Author Response · Authors · 2024-11-15
> > **Response to Reviewer pnDK (Part 3)**
> >
> > > Question 2: The underlying message of the paper is that context matters in determining whether an LLM should respond to a query. For instance, an LLM intended to train medical professionals should probably respond to medical questions. Why is it fair, then, to evaluate LLMs like GPT-4o-mini or Claude, which are intended to be used in the “layperson chatbot” context? I’m not sure I see why merely providing a “context” to these chatbots should actually alter their behavior. Given that a prevailing concern with these models is that users might employ deception to elicit responses from the models that developers do not desire (i.e., via jailbreaking or prompt-engineering)–shouldn’t these LLMs also decline to respond when the context is provided? The argument that these LLMs should respond when context is provided presumes that users will always be truthful when providing information in the prompt.
> >
> > Thank you for this thought-provoking question! We would like to clarify our underlying assumption of the context and experimental design.
> >
> > - We design the contexts in CASE-Bench such that they can be verified, e.g. the correct user and the recipient information can be obtained via the system administration. The "context" here is not a part of the prompt to the chatbot given by the user but can be provided by e.g. system administrator or regulations.
> > - In CASE-Bench, we are not asking the LLM to respond to a query given the context directly. Instead, we ask the LLM to judge if responding to the query is safe or not given the context while assuming the context is verified and reliable. This evaluation reflects whether the model understands the context and its influence on the safety of answering a user query from a judge's or administrator’s perspective. Hence the context is not altering the model behaviour to give the actual response, but altering the model's judgement assuming the context is true.
> > - That being said, our results showed that Claude is better at understanding the context and making better safety judgements under different contexts. This also indicates that, when Claude is tasked with verified context in a real scenario, it is more likely to succeed in providing information and suffer less from over-refusal problems compared to GPT-4o-mini when used off-the-shelf.
> >
> > In the revised paper, we have provided the explanation for the assumption in Section 3, and have included the explanation of these assumptions in the experimental design in Section 5.2 and the analysis of the results in Section 5.3.
> >
> > > Question 3: Could the authors share more information on the contexts?
> >
> > - As mentioned in our response to Weakness 2, we provide an example in Figure 1, and more examples are given in Appendix D-1 which we referred to in Section 1 in the revised paper. A link to the entire benchmark is given in the abstract.

---

> > > ### Comment · Reviewer_pnDK · 2024-11-21
> > >
> > > Thank you for the detailed responses! The additional detail is really helpful–I've adjusted my review accordingly.

---

> > > > ### Author Response · Authors · 2024-11-22
> > > > **Thank You for Your Support**
> > > >
> > > > Thank you for your positive feedback and for taking the time to review our responses. We’re grateful for your consideration and support!

---

### Author Response · Authors · 2024-11-29
**General Response to Reviewers**

Dear reviewers,

We would like to express our sincere gratitude for the time and effort you have dedicated to reviewing our work. Below, we would like to address your primary concerns for your convenience:

1. **Use of context and its validation**:
    - The contexts are meticulously designed to be verifiable. To achieve this, we carefully selected and structured CI parameters that can be validated through practical mechanisms. For example, these contexts can apply to systems with basic security measures, such as verifying the identity of the user interacting with the system. Within this framework, CASE-Bench primarily focuses on evaluating whether **given these contexts, the LLM can effectively make context-aware safety judgments**.

2. **Concerns about Jailbreaking**:
    - We would like to emphasize that the context in CASE-Bench is **separate** from the user prompt provided during interactions with the system. This separation can be practically implemented using methods like adapters or soft prompts, which can be trained to distinguish between verified context and user input during the RLHF stage such that jailbreaking via user-injected context can be easily avoided.
    - Moreover, there are cases where the context could actually aid in detecting jailbreak attempts. For instance, if a user falsely claims, "I am a professor in the [..] department," while the verified context indicates that the user is a high school student, this discrepancy can be identified and flagged.

**To reiterate our contributions and novelty**:

- **First to incorporate context in safety evaluation**: Our work investigates the critical observation that safety evaluations are often context-dependent. To ensure relevance and quality, we designed the contexts in CASE-Bench based on the well-established Contextual Integrity (CI) theory.
- **A novel context-aware benchmark with large-scale high-quality human annotations**: We use statistical methods to ensure its quality and scalability. The dataset also includes **non-binary** safety ratings, providing the foundation for future research on uncertainty in human judgments.

Once again, we deeply appreciate your thoughtful feedback and constructive suggestions. We hope that our responses have addressed your concerns satisfactorily. If there are any remaining questions or points of discussion, we would be delighted to engage further and clarify.


Sincerely,

Authors

---

### Meta-Review · Area_Chair_zqYU · 2024-12-21

**Metareview:**

This paper proposes CASE-Bench, a benchmark for evaluating LLMs' refusal accuracies for problematic queries. The reviewers agree the paper addresses a key issue in LLM safety evaluation and provides extensive experiments. However, after the rebuttal, there are remaining concerns on additional jailbreaking, how contexts can be verified/reliable, and query safety. There was an additional discussion among the reviewers, and the consensus is that there are still shared concerns about the contextuality framing and questions about jailbreaking and also some disagreement with the premise/setup of this work. In conclusion, I think the paper can be improved and should go through another round of reviewing.

**Additional Comments On Reviewer Discussion:**

Main remaining concerns:
- Jailbreaking: although the authors do clarify the jailbreaking problem, the reviewers do not seem fully convinced.
- Contextuality: it is still a bit unclear how the contexts can be verified and reliable.
- Query safety: discussing child laborers should be considered unsafe regardless of the context.

---

### Decision · Program_Chairs · 2025-01-22

Reject